# Separation of Five Flavonoids from Aerial Parts of *Salvia Miltiorrhiza* Bunge Using HSCCC and Their Antioxidant Activities

**DOI:** 10.3390/molecules24193448

**Published:** 2019-09-23

**Authors:** Fan Yang, Yingxue Qi, Wei Liu, Jia Li, Daijie Wang, Lei Fang, Yongqing Zhang

**Affiliations:** 1Key laboratory of Natural Pharmaceutical Chemistry, Shandong University of Traditional Chinese Medicine, Jinan 250200, China; yangfan65101@126.com (F.Y.); 13589120869@163.com (Y.Q.); LJYTL7172@163.com (J.L.); 2Key Laboratory of TCM Quality Control Technology, Shandong Analysis and Test Center, Qilu University of Technology (Shandong Academy of Sciences), Jinan 250014, China; Liuwei0074@163.com (W.L.); wangdaijie@126.com (D.W.); 3School of Biological Science and Technology, University of Jinan, Jinan 250200, China; fleiv@163.com

**Keywords:** aerial parts of *Salvia miltiorrhiza* Bunge, HSCCC, flavonoids, antioxidant activities

## Abstract

The aerial parts of *Salvia miltiorrhiza* Bunge, as the non-medicinal parts, are always discarded during harvesting, resulting in a huge waste of resources and environmental pressure. Due to the high flavonoid content and their antioxidant activities characteristics, the aerial parts of *S. miltiorrhiza* can be developed into natural antioxidants and used in foods. A high-speed counter-current chromatography (HSCCC) method, using a two-phase solvent system composed of *tert*-butyl methyl ether/*n*-butanol/acetonitrile/water (3:1:1:20, *v*/*v*), was the first to successfully isolate five flavonoids from the aerial parts of *S. miltiorrhiza* in one attempt, and separately categorized as rutin (**1**), isoquercitrin (**2**), kaempferol-3-*O*-α-l-rhamnopyranosyl-(1→6)-β-d-glucopyranoside (**3**), kaempferol-3-*O*-β-d-glucopyranoside (**4**) and apigenin-7-*O*-β-d-glucopyranoside (**5**) after identification. The purities of these plant isolates were 97.3%, 99.5%, 92.8%, 98.1% and 98.7%, respectively. All the flavonoids were identified by HR-ESI-MS, 1D and 2D NMR. Compounds **3** and **5** were firstly isolated from the plant of *S. miltiorrhiza.* Results from antioxidant assays showed that rutin (**1**) and isoquercitrin (**2**) had higher antioxidant capacities compared to L-ascorbic acid as the positive control.

## 1. Introduction

The dried roots and rhizomes of *Salvia miltiorrhiza* Bunge (SM), designated as Danshen in China or Tanshen in Japan, are widely distributed in both China and Japan [1]. As an extremely popular herb in China, SM is often used alone or in combination with other herbs. SM is also added to many traditional Chinese medicine preparations, such as Fufang Danshen tablets, Compound Danshen dripping pills, Danshen injection and Xiangdan injection [2]. In the aspect of pharmacological activities, SM has been used for the treatment of various diseases, including cerebrovascular disease [3], coronary heart disease [4], Parkinson’s disease [5], Alzheimer’s disease [6], renal deficiency [7], hepatocirrhosis [8,9], bone loss [10], and cancer [11].

Now the demand for SM is great, and its sources mainly depend on cultivation. The dried roots and rhizomes of SM are used as a medicine, while the aerial parts of SM are generally discarded during harvesting. This has resulted in serious waste of resources and environmental pressures. In view of this, it becomes necessary to study the chemical constituents and pharmacological activities of the aerial parts of SM. Studies show that the aerial parts of SM are rich in phenolic acids, flavonoids and triterpenoids, etc. [12,13,14,15]. They are proved to have some pharmacological activities, such as preventing and treating cardio-cerebral vascular disease [12] and chronic renal failure [16], improving the intestinal microecological environment [17] and antioxidant activity [2]. 

At present, more and more people have changed the focus of food additives from synthetics to natural antioxidants. Natural antioxidants have been proved to improve food quality and stability, as well as act as nutraceuticals to control free radical chain reactions in biological systems [18]. Flavonoids are the most important components in the aerial parts of SM, which have the potential to be developed into natural antioxidants because of the general antioxidant activity of flavonoids. To explore the antioxidant properties of flavonoids, there is a need to isolate these compounds. However, the availability on the reports on the isolation of flavonoids from the aerial parts of SM are, at present, very scarce.

The traditional methods used to isolate complex mixtures from SM frequently include silica gel chromatography, polyamide column chromatography, prep-HPLC or thin-layer chromatography (TLC) and others. But most of these methods may lead to long separation times, consumption of large amounts of solvents, a high risk of irreversible sample adsorption, and low sample recovery [19]. It is thus required to develop a fast and efficient method to isolate and purify flavonoids from aerial parts of SM. High-speed counter-current chromatography (HSCCC) is a liquid-liquid partition chromatographic technique that has been widely used in the isolation of natural products, with the advantages of high recovery, simple sample preparation, simple operation, low solvent consumption, high efficiency and ideal reproducibility [20]. However, as far as we know, there have been no reports on the isolation and separation of flavonoids by HSCCC from aerial parts of SM. In this paper, a new HSCCC method have been successfully developed to rapidly and efficiently isolate the five main flavonoids. The structures of the compounds were elucidated by ESI-MS combined with ^1^H and ^13^C-NMR. The chemical structures of the isolated flavonoids are shown in Figure 1.

## 2. Results and Discussion

### 2.1. Selection of the HSCCC Solvent Systems

For HSCCC separation, it is crucial to choose a solvent system with moderate *K* values for the target compounds. Higher *K* values may result in overly large peak breadth and extended elution time. However, lower *K* values may result in poor peak resolution. In this study, several solvent systems with different proportions were tested. The results were described in Table 1. When the solvent systems composed of ethyl acetate/*n*-butanol/water (4:1:5, *v*/*v*/*v*) was used, the *K* values were large, especially for compounds **3**, **4** and **5**. Thus, it became difficult to elute target compounds. Then, the solvent system composed of *n*-butanol/water (1:1, *v*/*v*) was tested, but was not suitable due to a lesser *K* value. Therefore, it was once again difficult to separate target compounds. Moreover, the solvent systems composed of *tert*-butyl methyl ether/*n*-butanol/acetonitrile/water with different proportions were tested. When the ratios of 2:2:1:5 (*v*/*v*/*v*/*v*) and 2.5:1.5:1:5 (*v*/*v*/*v*/*v*) were used, the *K* values of compounds **2**, **4** and **5** were large, which may lead to a long elution time. Then, the ratios of 3:1:1:5 and 3.5:0.5:1:5 were tested. As shown in Table 1, the *K* value of compound **1** in *tert*-butyl methyl ether/*n*-butanol/acetonitrile/water (3.5:0.5:1:5, *v*/*v*/*v*/*v*) was very small. When using *tert*-butyl methyl ether/*n*-butanol/acetonitrile/water (3:1:1:5, *v*/*v*/*v*/*v*), appropriate *K* values were obtained.

Considering the large consumption of the lower phase in the experiment, in order to conserve the solvent, the *K* value of the two-phase solvent system consisting of *tert*-butyl methyl ether/*n*-butanol/acetonitrile/water (3:1:1:20, *v*/*v*/*v*/*v*) was determined. The results demonstrated that, compared with the two-phase solvent system consisting of *tert*-butyl methyl ether/*n*-butanol/acetonitrile/water (3:1:1:5, *v*/*v*/*v*/*v*), the variation of the *K* value in this two-phase solvent system was insignificant. Therefore, the two-phase solvent system consisting of *tert*-butyl methyl ether/*n*-butanol/acetonitrile/water (3:1:1:20, *v*/*v*/*v*/*v*) was selected as the optimal solvent system for the experiment.

### 2.2. Selection of the Optimum HSCCC Experimental Conditions

Suitable experimental conditions of HSCCC can greatly shorten the isolation time and improve the isolation effect. Firstly, the retention volume of the upper phase of the *tert*-butyl methyl ether/*n*-butanol/acetonitrile/water (3:1:1:20, *v/v/v/v*) system was measured at different flow rates in Table 2. The results showed that the retention volume of the upper phase held on a high level (over 60%) at different flow rates. The flow rates have an insignificant effect for separation in this solvent system. Therefore, change in flow rate was a possible method for improving separation efficiency.

Then, in order to optimize an optimal experimental condition, many experiments were carried out to determine the influences of different flow rates on the isolation time and effect. In the first method, the lower phase was used for elution at a flow rate of 2.0 mL/min. After two of the target compounds had been eluted, the elution was continued at a flow rate of 4.0 mL/min. Two other target compounds were again eluted successively, after which the rotation rate was turned off and the last target compound was extruded from the column using the lower phase. As shown in Figure 2A, this method could effectively isolate these five target compounds, but the isolation time was too long, taking about 8 h.

Subsequently, the method was further modified by carrying out the elution at a flow rate of 4.0 mL/min from the beginning. After four target compounds had been eluted, the flow rate was turned off, and the last target compound in the column was ejected using the lower phase. As seen in Figure 2B, this method could shorten the isolation time to about 6.7 h, but the purity of compound **1** (peak I) was only 83.5%, and the time taken to reach the peaks for compounds **3** and **4** were still too slow.

Therefore, the method was modified again, with the elution firstly at a flow rate of 2.0 mL/min. When compound **1** (peak I, 24 mg) presented, the flow rate was changed to 10 mL/min. After compounds **3** (peak II, 19 mg), **2** (peak III, 30 mg) and **5** (peak IV, 14 mg) were eluted successively, the rotation rate was turned off and the solution in the column was ejected using the lower phase to obtain compound **4** (peak V, 12 mg). Figure 2C shows that this method can successfully shorten the time to 4.7 h, completely isolate all the compounds, and greatly improve the isolation efficiency and effect.

Figure 3 presents the HPLC chromatograms of the crude extracts and all the target compounds. It can be seen that the purities of compounds **1** to **5** reached 97.3%, 99.5%, 92.8%, 98.1% and 98.7% respectively, through calculating the proportion of the peak areas of target compounds in HPLC, suggesting that this method can successfully isolate compounds **1** to **5** from the aerial parts of SM.

### 2.3. Identification of Compounds

Compound **1** (Peak I in Figure 1, Appendix A): Yellow powder (CH_3_OH), ESI-MS *m/z* 609.0526 [M − H]^−^. ^1^H-NMR (DMSO-*d_6_*, 400 MHz) δ: 12.59 (1H, s), 7.56 (1H, dd, *J* = 9.1, 2.3 Hz, H-6′), 7.53 (1H, d, *J* = 2.1 Hz, H-2′), 6.84 (1H, d, *J* = 8.3 Hz, H-5′), 6.38 (1H, d, *J* = 2.0 Hz, H-8), 6.19 (1H, d, *J* = 2.1 Hz, H-6), 5.34 (1H, d, *J* = 7.8 Hz, H-1″), 4.38 (1H, s, H-1′′′), 0.99 (3H, d, *J* = 6.1 Hz, H-6′′′). ^13^C-NMR (DMSO-*d*_6_, 100 MHz) *δ*: 177.3 (C-4), 164.1 (C-7), 161.2 (C-5), 156.6 (C-9), 156.4 (C-2), 148.4 (C-4′), 144.8 (C-3′), 133.3 (C-3), 121.6 (C-6′), 121.2 (C-1′), 116.3 (C-5′), 115.2 (C-2′), 103.9 (C-10), 101.2 (C-1″), 100.7 (C-1′′′), 98.7 (C-6), 93.6 (C-8), 76.5 (C-3″), 75.9 (C-5″), 74.1 (C-2″), 71.9 (C-4′′′), 70.6 (C-3′′′), 70.4 (C-4″), 70.0 (C-2′′′), 68.2 (C-5′′′), 66.9 (C-6″), 17.7 (C-6′′′). Thus, the structure of **1** was defined as rutin by comparison of its MS, ^1^H-and ^13^C-NMR data with literature data [21].

Compound **2** (Peak III in Figure 1, Appendix A): yellow powder (CH_3_OH), ESI-MS *m/z* 463.0160 [M − H]^−^. ^1^H-NMR (400 MHz, DMSO-*d*_6_) δ: 12.64 (1H, s, 5-OH), 10.78 (1H, s, 7-OH), 9.29 (2H, overlapped, 3′, 4′-OH), 7.58 (2H, overlapped, H-2′, 6′), 6.84 (1H, d, *J* = 9.0 Hz, H-5′), 6.40 (1H, d, *J* = 2.0 Hz, H-8), 6.20 (1H, d, *J* = 2.0 Hz, H-6), 5.46 (1H, d, *J* = 7.0 Hz, GlcH-1″). ^13^C-NMR (100 MHz, DMSO-*d*_6_) *δ*: 177.4 (C-4), 164.2 (C-7), 161.2 (C-5), 156.3 (C-2), 156.2 (C-9), 148.5 (C-4′), 144.8 (C-3′), 133.2 (C-3), 121.6 (C-6′), 121.2 (C-1′), 116.2 (C-5′), 115.2 (C-2′), 104.0 (C-10), 100.9 (C-1″), 98.7 (C-6), 93.5 (C-8), 77.6 (C-5″), 76.5 (C-3″), 74.1 (C-2″), 69.9 (C-4″), 61.0 (C-6″). Thus, the structure of **2** was defined as isoquercitrin by comparison of its MS, ^1^H-and ^13^C-NMR data with literature data [22].

Compound **3** (Peak II in Figure 1, Appendix A): yellow powder (CH_3_OH), ESI-MS *m/z* 593.0571 [M − H]^−^. ^1^H-NMR (400 MHz, DMSO-*d*_6_) δ: 7.98 (2H, d, *J* = 8.5 Hz, H-2′, 6′), 6.87 (2H, d, *J* = 8.4 Hz, H-3′, 5′), 6.39 (1H, d, *J* = 2.0 Hz, H-8), 6.18 (1H, d, *J* = 2.0 Hz, H-6), 5.30 (1H, d, *J* = 7.4 Hz, H-1″), 4.37 (1H, s, H-1′′′), 0.98 (3H, d, *J* = 6.1 Hz, H-6′′′). ^13^C-NMR (100 MHz, DMSO-*d*_6_) *δ*: 177.2 (C-4), 161.1 (C-5), 160.0 (C-4′), 156.7 (C-9), 156.5 (C-2), 133.1 (C-3), 130.8 (C-2′, 6′), 120.8 (C-1′), 115.1 (C-3′, 5′), 103.7 (C-10), 101.5 (C-1″), 100.8 (C-1′′′), 99.0 (C-6), 93.9 (C-8), 76.4 (C-3″), 75.7 (C-5″), 74.1 (C-2″), 71.8 (C-4′′′), 70.6 (C-3′′′), 70.3 (C-2′′′), 69.9 (C-4″), 68.2 (C-5′′′), 66.8 (C-6″), 17.7 (C-6′′′). Thus, the structure of **3** was defined as kaempferol-3-*O*-α-l-rhamnopyranosyl-(1→6)-β-d-glucopyranoside by comparison of its MS, ^1^H- and ^13^C-NMR data with literature data [23].

Compound **4** (Peak V in Figure 1, Appendix A): yellow powder (CH_3_OH), ESI-MS *m/z* 447.0218 [M − H]^−^. ^1^H-NMR (400 MHz, DMSO-*d*_6_) δ: 12.62 (1H, s, 5-OH), 8.05 (2H, d, *J* =8.6 Hz, H-2′, 6′), 6.88 (2H, d, *J* =8.8 Hz, H-3′, 5′), 6.43 (1H, d, *J* = 2.1 Hz, H-8), 6.20 (1H, d, *J* = 2.0 Hz, H-6), 5.47(1H, d, *J* = 7.3, H-1′′), 3.75-3.08 (6H, overlapped, H-2′′-6′′). ^13^C-NMR (100 MHz, DMSO-*d*_6_) *δ*: 177.5 (C-4), 164.4 (C-7), 161.2 (C-5), 160.0 (C-4′), 156.4 (C-9), 156.2 (C-2), 133.2 (C-3), 130.9 (C-2′, 6′), 124.0 (C-1′), 115.2 (C-3′, 5′), 104.0 (C-10), 101.6 (C-1″), 99.1 (C-6), 98.8 (C-8), 77.5 (C-3″), 76.6 (C-5″), 74.2 (C-2″), 70.2 (C-4″), 61.1 (C-6″). Thus, the structure of **4** was defined as kaempferol-3-*O*-β-d-glucopyranoside by comparison of its MS, ^1^H- and ^13^C-NMR data with literature data [24].

Compound **5** (Peak IV in Figure 1, Appendix A): yellow powder (CH_3_OH), ESI-MS *m/z* 431.0825 [M–H]^−^. ^1^H-NMR(DMSO-*d_6_*, 400 MHz) δ: 13.00 (1H, s, 5-OH), 7.95 (2H, d, *J* = 8.4 Hz, H-2′, 6′), 6.95 (2H, t, *J* = 8.8 Hz, H-3′, 5′), 6.87 (1H, s, H-3), 6.84 (1H, s, H-8), 6.44(1H, s, H-6), 5.06 (1H, d, *J* = 7.4 Hz, H-1′′), 3.72~3.19 (6H, overlapped, H-2′′-6′′). ^13^C-NMR (DMSO-*d_6_*, 100 MHz) *δ*: 182.0 (C-4), 164.4 (C-2), 163.0 (C-7), 161.8 (C-5), 161.1 (C-4′), 157.0 (C-9), 128.6 (C-2′, 6′), 120.7 (C-1′), 116.1 (C-3′, 5′), 105.3 (C-10), 103.0 (C-3), 99.9 (C-6), 99.5 (C-1′′), 94.8 (C-8), 77.2 (C-3′′), 76.5(C-5′′), 73.1 (C-2′′), 69.6 (C-4′′), 60.6 (C-6′′). Thus, the structure of **5** was defined as apigenin-7-*O*-β-d-glucopyranoside by comparison of its MS, ^1^H- and ^13^C-NMR data with literature data [22].

### 2.4. Antioxidant Activity

IC_50_ value is a significant standard for antioxidant activity. The smaller the IC_50_ value, the better the antioxidant activity. As shown in Table 3, every sample exhibited different antioxidant activity. The antioxidant activities on DPPH% of the crude extract and isolated compounds were found to be in the order of rutin (**1**) > isoquercitrin (**2**) > crude flavonoid extract > kaempferol-3-*O*-α-l-rhamnopyranosyl-(1→6)-β-d-glucopyranoside (**3**) > kaempferol-3-*O*-β-d-glucopyranoside (**4**) > apigenin-7-*O*-β-d-glucopyranoside (**5**), with IC_50_ values ranging from 16.7 to 177.1 µg/mL.

## 3. Materials and Methods

### 3.1. Reagents and Materials

The ethanol, ethyl acetate, methanol, chloroform, *tert*-butyl methyl ether, *n*-butyl alcohol and petroleum ether (60–90 °C) used for the preparation of crude extract and counter-current chromatography (CCC) separations were analytical grade (Sinopharm Chemical Reagent Co., Ltd., Shanghai, China). HPLC-grade methanol and acetonitrile were purchased from the Fisher Company (Fairlawn, NJ, USA). The water used was deionized by an osmosis Milli-Q system (Millipore, Bedford, MA, USA). Reverse osmosis Milli-Q water (Millipore, Bedford, MA, USA) was used.

The aerial parts of SM were obtained from a Chinese traditional medicine plantation in Tai′ an (Shandong, China) and identified by Dr. Jia Li (College of Pharmacy, Shandong University of Traditional Chinese Medicine). A voucher specimen (2018090805) was deposited at Shandong Analysis and Test Center.

### 3.2. Apparatus

The HSCCC equipment was TBE-300C (Shanghai, Tauto Biotech, China) with three multilayer coil separation columns of 300 mL (diameter of the PTFE tube as 2.6 mm) as well as a 20 mL manual sample loop. The HSCCC apparatus was equipped with four other instrument modules, including a TBP-5002 constant-flow pump (Tauto Biotechnique, Shanghai, China), a 8823A-UV Monitor at 254 nm (Beijing101 Emilion Technology, Beijing, China), a Model 3057 portable recorder (Yokogawa, Sichuan Instrument Factory, Sichuan, China), and a DC-0506 low constant temperature bath (Tauto Biotechnique, Shanghai, China) to maintain the temperature at 25 °C. HPLC separation was performed on a 1120 LC system (Agilent Technologies, Santa Clara, CA, USA) consisting of a quaternary pump, an online degasser, a diode array detector, an auto plate-sampler and a thermostatically controlled column compartment.

### 3.3. Preparation of Crude Extract

The dried aerial parts of SM (1.0 kg) were extracted with 80% ethanol (10.0 L) at room temperature. After being concentrated in vacuo, the ethanol extract (0.15 kg) was suspended in water and extracted with ethyl acetate (3.0 L) and *n*-butanol (3.0 L).

The *n*-butanol soluble fraction (32.3 g) was first separated over a macroporous adsorbent resin column. The column was eluted with water-ethanol (100:0, 70:30, 50:50, 30:70, 3:97, *v*/*v*), yielding five fractions. Then, the 50% ethanol eluate (17.4 g) was ready to be separated by HSCCC.

### 3.4. Selection of Solvent System

The selection of the solvent system is based on the partition coefficient (*K*-values). The *K*-values of the target compounds in the crude flavonoid extract from the aerial parts of SM were determined by HPLC. Five milliliters of each phase of the equilibrated two-phase solvent system were added to approximately 10 mg of the crude flavonoid extract and were shaken vigorously for 1 min. After the phases had fully separated, 1 mL of each layer was removed and dried in a stream of nitrogen. The residues were dissolved in 1 mL of methanol and analyzed by HPLC. The *K*-values of the target compounds were calculated according to the equation *K* = A_U_/A_L_, where A_U_ and A_L_ were the peak areas of target compound in the upper and lower phases, respectively.

### 3.5. Preparation of the Solvent System and Sample Solutions

For HSCCC separation, a two-phase solvent system, consisting of *tert*-butyl methyl ether/*n*-butanol/acetonitrile/water (3:1:1:20, *v*/*v*/*v*/*v*), was placed into a separating funnel. After shaking vigorously, the solution was allowed to stand for several minutes and was separated into two phases for the experiment. The upper phase was the stationary phase, while the lower was the mobile phase. For this HSCCC separation, 200 mg of crude flavonoid extract was dissolved in 20 mL isometric upper and lower phase.

### 3.6. Optimization of HSCCC Separation Procedure

Firstly, in order to select the optimal flow rate, the retention volume of the upper phase was measured at different flow rates. After the CCC column was filled with the upper phase, the rotation speed was set to 800 rpm, and then the lower phase was used at the flow rate of 2, 4, 6, 8, 10, 12, 14 mL/min successively to achieve equilibrium. The eluent was drained in a measuring cylinder. When the volume of the outflowing upper phase no longer changed, it was regarded that the equilibrium had been achieved.

The whole separation procedure had two stages: collection and extrusion. In the first stage, the six-way valve was in the position for collection. This stage encompassed establishment of hydrodynamic equilibrium, sample solution loading, and sample eluent collection. The CCC column was first completely filled with the upper phase at 30 mL/min in head-to-tail elution mode. The lower phase was pumped into the head of the CCC column at 2.0 mL/min, during which the apparatus was rotated at 800 rpm in a clockwise manner. The separation temperature was set at 25 °C. The effluents were continuously monitored at 254 nm by means of a portable recorder. After equilibrium had been reached, the crude flavonoid extract solution was injected using the manual sample loop. After the first target peak was collected, the flow rate was changed to 10 mL/min, while keeping the rotation speed unchanged, and elution was continued. Once the 2nd, 3rd, and 4th target peaks could be collected successively, the flow was stopped, immediately followed by the turning off the rotation speed. The lower phase was used to extrude the solution from the column at a flow rate of 10 mL/min.

### 3.7. HPLC Analysis

HPLC analyses of the extract and CCC fractions were performed on Agilent 1120 HPLC equipment with a C18 column (Waters symmetry, 5 μm, 4.6 mm × 250 mm, i.d.). The mobile phase was acetonitrile and 0.1% aqueous solution of formic acid (21:79, *v*/*v*) with a flowrate of 1.0 mL/min and a wavelength of 254 nm.

### 3.8. Structural Identification

The separated compounds were identified by ESI-MS, ^1^H-and ^13^C-NMR spectra. The ESI-MS experiment was performed on an Agilent 6520 Q-TOF (Agilent, Santa Clara, CA, USA). NMR spectra were performed on a Bruker AV-400 spectrometer (Bruker BioSpin, Rheinstetten, Germany) with DMSO-*d_6_* as solvent and chemical shifts (δ) being expressed in parts per million (ppm) coupled with the constant (*J*) in Hz.

### 3.9. Evaluation of Antioxidant Activity

DPPH (12.5 mg) was dissolved in 250 mL ethanol to make a standard solution at a concentration of 0.05 mg/mL. Serial dilution was used to provide standard solutions at concentrations of 0, 5, 10, 15, 20 and 25 μg/mL in ethanol. The absorbance values of the six solutions were measured by ultraviolet spectrophotometry, at 517 nm, to prepare the standard curve.

## 4. Conclusions

In conclusion, a new HSCCC method, using a two-phase solvent system composed of *tert*-butyl methyl ether/*n*-butanol/acetonitrile/water (3:1:1:20, *v*/*v*), was successfully established to isolate flavonoids from aerial parts of SM. Five flavonoids were obtained in one step, and their purities were 97.3%, 92.8%, 99.5%, 98.7% and 98.1%, respectively. Compounds **3** and **5** were firstly isolated from the plant of *S. miltiorrhiza*. There was no obvious difference in adding excess water in the two-phase solvent system, which could save a lot of reagents. Antioxidant assays results showed that rutin (**1**) and isoquercitrin (**2**) have stronger antioxidant capacities as compared to l-ascorbic acid as the positive control.

## Figures and Tables

**Figure 1 molecules-24-03448-f001:**
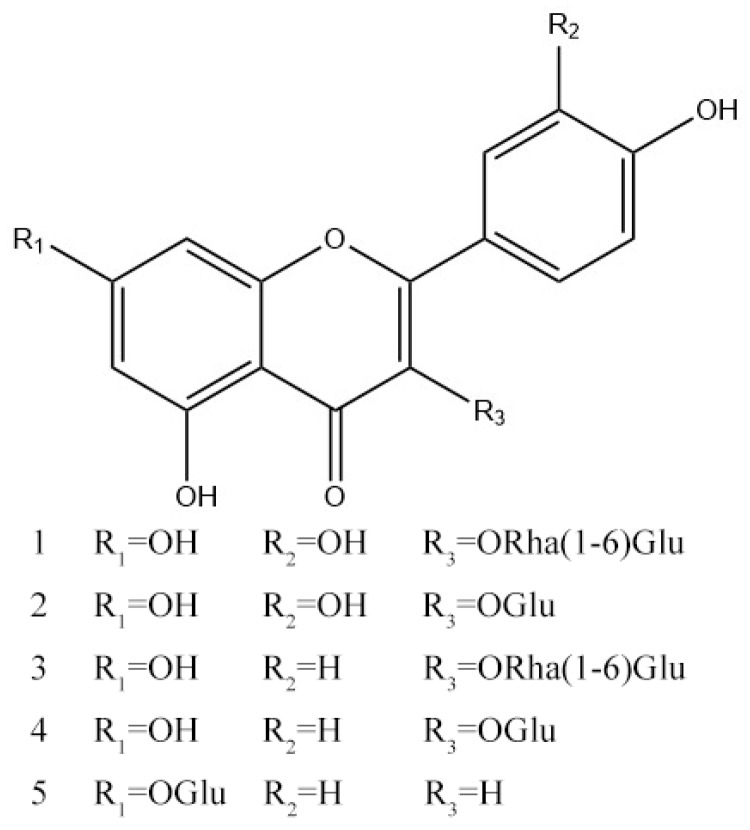
Chemical structures of compounds **1**–**5**.

**Figure 2 molecules-24-03448-f002:**
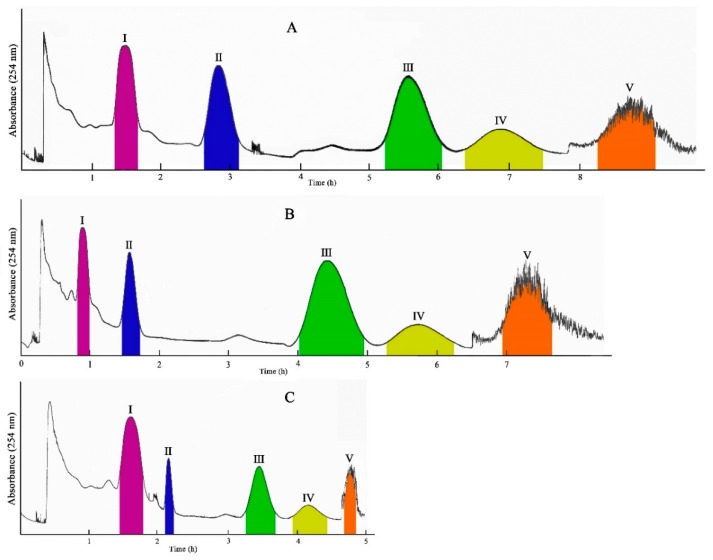
HSCCC chromatogram of the crude sample from aerial parts of *Salvia miltiorrhiza* Bunge. Flow rate: (**A**) 2.0→4.0 mL/min, (**B**) 4.0→10.0 mL/min, (**C**) 2.0→10.0 mL/min; stationary phase: upper phase; mobile phase: lower phase; revolution speed: 800 rpm; detection wavelength: 254 nm; separation temperature: 25 °C.

**Figure 3 molecules-24-03448-f003:**
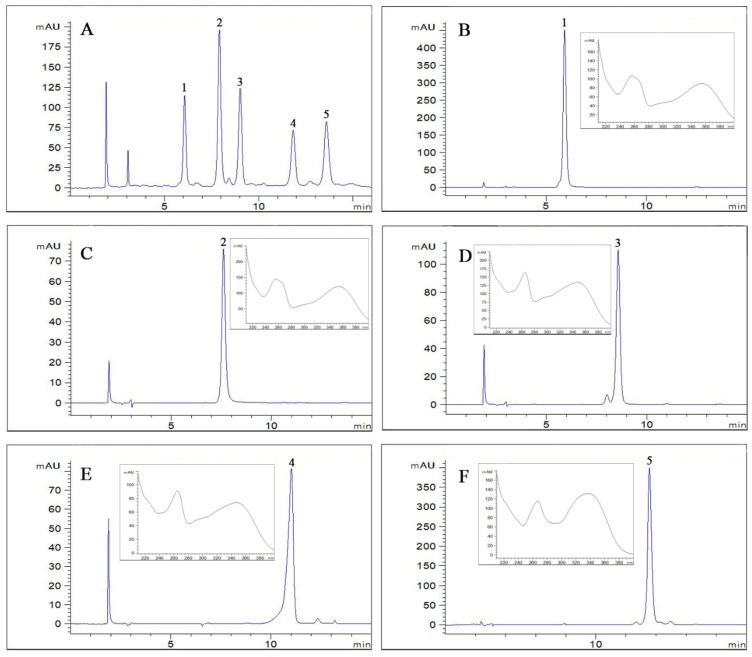
(**A**) HPLC analysis of the crude sample; (**B**–**F**) HPLC chromatograms and UV spectra of compounds **1**–**5**. Column: C18 column (Waters symmetry, 5 μm, 4.6 mm × 250 mm, i.d.); mobile phase: acetonitrile (**A**) 0.1% aqueous solution of formic acid (B) (0–15 min, 21:79, *v*/*v*); flow rate: 1.0 mL/min; detection wavelength: 254 nm; column temperature: 25 °C.

**Table 1 molecules-24-03448-t001:** The *K* values of target compounds in HSCCC separation with different solvent systems.

Solvent systems	Ratio (*v*/*v*)	*K*				
		1	2	3	4	5
ethyl acetate/*n*-butanol/water	4:1:5	0.67	5.61	1.92	52.31	18.64
*n*-butanol/water	1:1	0.42	0.67	0.61	0.75	0.85
*tert*-butyl methyl ether/*n*-butanol/acetonitrile/water	2:2:1:5	1.83	7.25	3.70	15.92	10.06
*tert*-butyl methyl ether/*n*-butanol/acetonitrile/water	2.5:1.5:1:5	1.44	7.68	2.95	12.70	8.11
***tert*-butyl methyl ether/*n*-butanol/acetonitrile/water**	**3:1:1:5**	**0.62**	**4.68**	**1.51**	**9.53**	**5.78**
*tert*-butyl methyl ether/*n*-butanol/acetonitrile/water	3.5:0.5:1:5	0.11	2.22	0.57	4.39	2.53
***tert*-butyl methyl ether/*n*-butanol/acetonitrile/water**	**3:1:1:20**	**0.59**	**5.11**	**1.42**	**12.65**	**6.35**

In general, the upper-phase and lower-phase are prepared in a ratio of 1:1 for HSCCC separation. All of solvent systems tested are based on the standard except the 3:1:1:20 solvent system. The *K* value of *tert*-butyl methyl ether/*n*-butanol/acetonitrile/water (3:1:1:5, *v/v/v/v*) is the most appropriate in these solvent systems. The ratio of 3:1:1:20 is just an optimization based on the ratio of 3:1:1:5 for saving reagents, nonetheless, the ratios of 3:1:1:5 and 3:1:1:20 are equally appropriate in the aspect of *K* value.

**Table 2 molecules-24-03448-t002:** The retention volume of the upper phase at different flow rates.

Flow Rate (mL/min)	Retention volume (L)	Ratio (Retention volume/Total volume, %)
2	243	81.0
4	225	75.0
6	215	71.7
8	205	68.3
10	200	66.7
12	193	64.3
14	188	62.7

**Table 3 molecules-24-03448-t003:** Antioxidant activities of compounds **1**–**5**.

Samples	DPPH (IC_50_, μg/mL) ^a^
Crude flavonoid extract	34.2
1	16.7
2	17.1
3	136.6
4	159.5
5	177.1
l-Ascorbic acid ^b^	6.8

^a^ Each value is presented as mean ± SD (n = 3); ^b^ Compared as control.

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
