# Peer review of "Separation of Five Flavonoids from Aerial Parts of Salvia Miltiorrhiza Bunge Using HSCCC and Their Antioxidant Activities"

_molecules, 2019, doi:10.3390/molecules24193448_

Round 1

Reviewer 1 Report

This paper describes the countercurrent chromatographic (CCC) separation of five flavonoids extracted from aerial parts of Salvia miltiorrhiza Bunge and their antioxidant activities. The separation was successfully achieved using a two-phase solvent system composed of tert-butyl methyl ether/1-butanol/acetonitrile/water (3 : 1 : 1 : 20, v/v). The isolated compounds were identified as rutin (1), isoquercitrin (2), kaempferol-3-O-α-L-rhamnopyranosyl-(1→6)-β-D-glucopyranoside (3), kaempferol-3-O-β-D-glucopyranoside (4) and apigenin-7-O-β-D-glucopyranoside (5), respectively, using the instrumental analysis with high purity. The anitioxidant acitivities were also measured while these compounds were already-known compounds. The paper is well organized so that it is recommended to the publication in Molecules. Several comments for the correction are described below:

page 1, line 11 – 12: Correspondence, e-mail address, telephone number should be clearly described.

page 1, line 19: tert-butyl methyl ether/n-butanol/acetonitrile-water ---> tert-butyl methyl ether/n-butanol/acetonitrile/water

page 2, line 56: pre-HPLC ---> prep-HPLC

page 3, line 98: the retention volume of the upper phase was measured at different flow rates in Table 2. ---> the retention volume of the upper phase of the tert-butyl methyl ether/n-butanol/acetonitrile/water (3 : 1 : 1: 20, v/v/v/v) system was measured at different flow rates in Table 2 (The two-phase solvent system used for the measurement of the retention volume should be clearly described in this section).

page 5, Figure 3: Upper chromatogram of the right side: add alphabet: B

page 5, line 132: Figure 3. presents ---> Figure 3 (not bold, delete a period) presents

page 5, line 137, 146, page 6, line 154, 163, 171: yellow powder (MeOH): the meaning of “(MeOH)” should be clearly described.

page 7, line 214: n-butanol ---> n (italic)-butanol

page 8, line 261: DMSO ---> DMSO-d6

Author Response

I accepted all your suggestions and have modified the manuscript according to your suggestions. Thank you for your suggestions.

Reviewer 2 Report

Manuscript (molecules-595010) entitled „ Separation of five flavonoids from aerial parts of Salvia miltiorrhiza Bunge using HSCCC and their antioxidant activities “ was aimed to develop new HSCCC method to rapidly and efficiently isolate the five main flavonoids of Salvia miltiorrhiza Bunge, very popular in traditional Chinese medicine preparations. From experimentation to data evaluation, everything is well organized and clearly described and the HR-ESI-MS and NMR analysis appears to be carefully performed. In my opinion, the quality of this manuscript is acceptable to be published in Molecules, after minor revision. Some remarks are summarized as follows:

Line 24 and 271           This is about flavonoid glycosides, not iridoid glycosides.

Figure 2c        Only 5 compounds were isolated in this plant; this seems a very small number. The chromatograms show that they are the most represented, but what about the peaks at about 1.3h, 2h and 3h (Fig. 2c)? Could those smaller peaks somehow be identified?

Figure 3a        What are the peaks at 2 min and 3 min?

Table 3           If data for IC50 purity of standard compounds can be found in the literature, then they should be compared with the experimentally obtained values. Antioxidant activity for crude extract and for each compound is determined herein. I think this section needs to be expanded with discussion why the IC50 of individual compounds is in some cases smaller and larger in some than the IC50 of crude extract.

Author Response

Line 24 and 271: The errors have been corrected.

Figure 2c: This 5 compounds were isolated from a part of n-butyl alcohol extract of Salvia miltiorrhiza Bunge, not form  the plant. In addition, the smaller peaks were also measured by HPLC, but the results showed that the purities of these peaks were not very high. So these smaller peaks were not added in this paper. Then we will collect these smaller peaks and use other methods to isolate them.

Figure 3a: The peaks is the solvent of samples.

Table 3 : I have added some sentences(Line 180) to explain the meaning of the IC50 values.